

# An approach for semantic integration of heterogeneous data sources

Giuseppe Fusco and Lerina Aversano

Department of Engineering, University of Sannio, Benevento, BN, Italia

## ABSTRACT

Integrating data from multiple heterogeneous data sources entails dealing with data distributed among heterogeneous information sources, which can be structured, semi-structured or unstructured, and providing the user with a unified view of these data. Thus, in general, gathering information is challenging, and one of the main reasons is that data sources are designed to support specific applications. Very often their structure is unknown to the large part of users. Moreover, the stored data is often redundant, mixed with information only needed to support enterprise processes, and incomplete with respect to the business domain. Collecting, integrating, reconciling and efficiently extracting information from heterogeneous and autonomous data sources is regarded as a major challenge. In this paper, we present an approach for the semantic integration of heterogeneous data sources, DIF (Data Integration Framework), and a software prototype to support all aspects of a complex data integration process. The proposed approach is an ontology-based generalization of both Global-as-View and Local-as-View approaches. In particular, to overcome problems due to semantic heterogeneity and to support interoperability with external systems, ontologies are used as a conceptual schema to represent both data sources to be integrated and the global view.

## INTRODUCTION

The large availability of data within the enterprise context and even in any inter-enterprise context, the problem arises of managing information sources that do not use the same technology, do not have the same data representation, or that have not been designed according to the same approach. Thus, in general, gathering information is a hard task, and one of the main reasons is that data sources are designed to support specific applications. Very often their structure are unknown to the large part of users. Moreover, the stored data is often redundant, mixed with information only needed to support enterprise processes, and incomplete with respect to the business domain. Collecting, integrating, reconciling and efficiently extracting information from heterogeneous and autonomous data sources is regarded as a major challenge. Over the years, several data integration solutions have been proposed:

- Distributed databases can be considered the first attempt to integrate databases. Data, instead of being stored on a single machine, is stored on different machines. Compared to the centralized case, database schema are more complicated by the need to physically

Corresponding author
Lerina Aversano,
aversano@unisannio.it

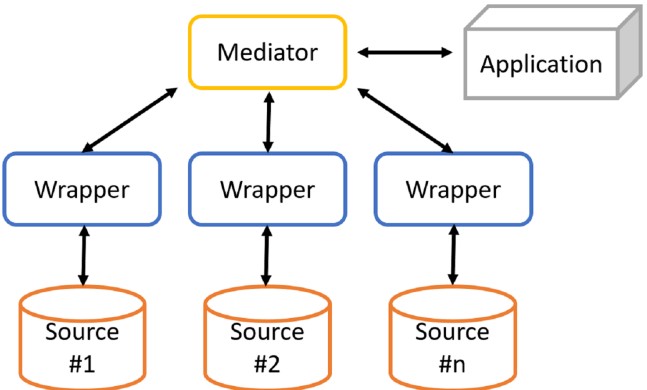

**Figure 1  Architecture of a generic mediation system.**

distribute data over multiple machines. Distributed databases require the complete integration of existing systems into a single homogeneous database. This is difficult to achieve due to technical issues (prohibitive conversion costs) and organizational difficulties (existing DBMSs belong to different organizations).

- Federated databases have been proposed to address these limits. They are a set of multiple independent sources each of which can exchange information with the others. A connection is established for each pair of sources, and such architecture is particularly suitable when communications in the system occur predominantly between pairs of sources.

The solution often adopted consists of the cooperative information systems (Fig. 1), in which there are two types of components: mediator and wrapper. The mediator coordinates the data flow between local sources and user applications. The mediator is not responsible for storing data, since it only stores a virtual and global view of real data (or global schema) and the mappings between the global and local views. In this way, applications will run queries over the virtual view. It will then be the mediator to build queries for individual sources of information. Instead, wrappers are software components that interact directly with their respective local sources as follows:

- to translate the conceptual schema of the local source into a global language;
- to submit queries to local sources;
- to retrieve results by sending them to the mediator, which will provide the user with a unified result.

This approach allows provide users with a unified interface (called mediated schema or global schema or global view) of sources, freeing them from manually managing each source. The open research problem is the need of a not statically constructed mediator, but the need of querying mediator responsible of accessing heterogeneous and dynamic data sources trough a global view without integrating or migrating the local data source. To overcome this research problem, this paper proposes an ontology based framework to support the analysis, acquisition and processing of data from heterogeneous sources, Data

Integration Framework (DIF). It exploits domain ontology and provides a generalization of both global view and local view approaches, based on data virtualization. The proposed framework addresses this issue by providing access to the data layer, consisting of autonomous data sources (e.g., DBs, spreadsheets), through the mediation of a global domain view, given in terms of an ontology, and the use of a semiautomatic mapping between the data layer and the ontology. Users do not have to know details of the data sources and can express their information needs as queries over the conceptual domain model. The proposes framework uses the ontology and mapping to reformulate the user queries into standard DB queries that are executed directly by the database management systems (DBMSs) of the sources. The global view provides a unified view of real data, so that applications and users who use data will have the perception of accessing a single data source rather than multiple sources. In this context, the work faced aspects of acquisition, integration and processing of heterogeneous data sources.

The paper is organized as follows. 'Aspects of a Data Integration Process', 'Related Work' presents, respectively, problems that characterize data integration and proposed solutions in the state of the art. 'Data Integration Framework' presents in detail the approach and architecture of the software system developed to support the integration of heterogeneous sources. 'DIF Supporting Tool' presents the DIF tool design and the main algorithms implemented. 'Case Study' presents a case study in order to show the results of the proposed solution. Finally, 'Conclusion' concludes this paper by submitting concluding remarks and mentioning some research issues related to data integration that are not addressed in this paper.

## ASPECTS OF A DATA INTEGRATION PROCESS

Data integration systems using the mediation approach are characterized by an architecture (Fig. 1) based on a global schema and a set of sources schema. The sources contain real data, while the global scheme provides a unified, integrated and reconciled view of local sources. The main components are: the mediator, which coordinates data flow between local sources and user applications, and wrappers, which directly interact with their respective local sources. Designing a data integration system is a complex task, which involves dealing with different issues.

The first issue is related to the heterogeneity of the sources, as sources adopt different models and data storage systems. This poses problems in defining the schema/global view. The purpose is to provide a view with an appropriate level of abstraction for all data in the sources.

The second issue is how to define mappings between global schema and local sources: in literature, in order to model the correspondences between the schemes, different approaches (*Lenzerini, 2002*) have been proposed as global-as-view and local-as-view. With global-as-view (GaV) approach the global schema is expressed in terms of views over data sources. With local-as-view (LaV) approach the global schema is specified independently of the sources, and the relationships between global schema and sources are established by defining each source as a view built over the global schema. Differences

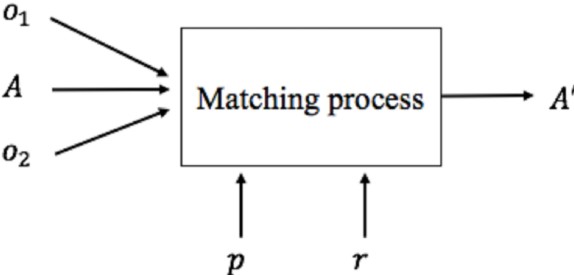

**Figure 2   Matching process.**

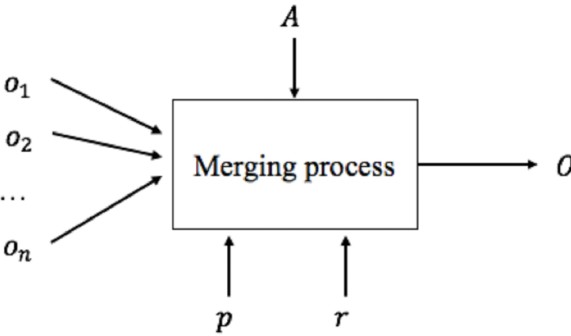

**Figure 3   Merging process.**

between the two approaches are discussed in *Lenzerini (2002)*. In order to overcome the limits of GaV and LaV approaches, techniques that combine the benefits of these approaches have also been proposed, mitigating their disadvantages in order to provide an alternative to data integration that is more flexible and scalable. The most interesting techniques are GLaV (Global and Local as View) (*Katsis & Papakonstantinou, 2009*; *Arens et al., 1993*) and BGLaV (BYU Global Local as View) (*Xu & Embley, 2004*).

Once the mapping approach is defined, it is necessary to define the methods and techniques to be used to generate mappings between the global and the local schema. This activity is called Schema Matching. The set of mappings is called alignment. A matching process (*Shvaiko & Euzenat, 2005*) (Fig. 2) defines an alignment ($A^{'}$) for each pair of schemas ($o_1, o_2$), making use of input parameters $p$ if necessary (for example, thresholds, weights), a previously generated input alignment ($A$) and additional external resources $r$.

We can now generate the global schema based on mappings defined in the schema matching activity. This activity is called Schema Merging. A merging process (Fig. 3) consists of integrating several existing schemes ($o_1, o_2, \ldots, o_n$) into a single global schema ($O$) based on the correspondences generated by the schema matching process $A$, any input parameters $p$ and external resources $r$. Different techniques and methodologies about schema merging have been proposed in the literature (*Lee & Ling, 2003*; *Fong et al., 2000*; *Chiticariu, Kolaitis & Popa, 2008*; *Kong, Hwang & Kim, 2005*).

**Table 1  Comparison between GaV and LaV.**

|  | GaV | LaV |
|---|---|---|
| Mapping | Global schema expressed in terms of views over data sources | Data sources expressed in terms of views over global schema |
| Query processing | Query unfolding | Query rewriting/Query answering |
| Global schema quality | Exact or Sound | Complete or Exact |
| Management effort | High: data source changes affect the global schema and other sources | Low: data source changes only impact the global schema |

Another issue is related to data storage: compared to managed data there are two approaches, called materialization and virtualization. With materialization, data is also present in the global schema. On the opposite, in the virtualization approach, data that resides in sources is only available when query processing activity is executed.

Once we merged local views into one unified global view, we can process a query posed over the global schema. This activity is called Query Processing, that is how to express it in terms of a set of queries that can be processed by the sources acquired. In the LaV approach the proposed solutions consist of query rewriting (or view-based query rewriting) and query answering (or view-based query answering). In the GAV approach query unfolding techniques are used. The differences between the two approaches are discussed in *Lenzerini (2002)*.

Once the query processing activity is performed, data from different sources need to be interpreted, that is, transformed into a common representation. Therefore, they must be converted, reconciled and combined.

Table 1 summarizes the approaches used in mappings definition between the global schema and local ones.

Based on the comparison approaches in Table 1 it is possible to observe that:

- The LaV approach involves a priori presence of a global schema, which must be well-built in terms of concepts and relationships between them. If it is not well-built, or the integrated schemas differ greatly from each other, the global schema must be continually modified, also taking into account the previously integrated data sources. If not, the changes affect only the global schema.
- With GaV approach, the global schema is incrementally built: it is modified every time a new data source is integrated, adding and/or modifying concepts and relationships based on current and previously integrated data sources. Conversely, in the LaV case, changes do not impact previous integrated data sources (if the overall schema is well-built).
- The LaV approach offers greater scalability when the number of integrated data sources increases, but when that number is relatively small, and the global schema is not well-built, the GaV approach increases the quality of global schema.

Moreover, in the context of semantic integration, the hybrid approach is surely the best solution but it reduces the reuse of local ontologies, since they have to refer to a common vocabulary. Therefore, considering a possible future reuse of local the ontologies, it is

possible to combine the presented approaches differently in order to support different cases and to present a data integration approach in order to provide different solutions as needed. The proposed approach, called DIF is based on these observations and seeks to combine the GaV and LaV approaches, exploiting ontologies to reach the goals.

## RELATED WORK

Several systems, methodologies and approaches have been proposed in literature to support data integration from heterogeneous sources, also based on ontologies (*Calvanese, Lembo & Lenzerini, 2001*).

To overcome problems due to semantic heterogeneity, it is useful to use ontologies (*Wache et al., 2001*). Depending on how ontologies are used, data integration systems can adopt different approaches, such as single ontology (adopted in SIMS (*Arens et al., 1993*; *Arens, Hsu & Knoblock, 1996*)), multiple ontology (adopted in OBSERVER (*Mena et al., 1996b*; *Mena et al., 1996a*)) and hybrid (adopted in KRAFT (*Preece, Hui & Gray, 1999*; *Preece et al., 2000*) and COIN (*Goh et al., 1999*)).

More recently in *Civili et al. (2013)* it is proposed Mastro Studio, a Java tool for ontology-based data access (OBDA). Mastro manages OBDA systems in which the ontology is specified in a logic specifically tailored to ontology-based data access and is connected to external data management systems through semantic mappings that associate SQL queries over the external data to the elements of the ontology.

TSIMMIS (The Stanford-IBM Manager of Multiple Information Sources) (*Chawathe et al., 1994*) is based on an architecture that exposes a wrapper hierarchy (called Translators) and mediators.

TSIMMIS approach is global-as-view. Wrappers convert data to a common data model called OEM (Object Exchange Model) and mediators combine and integrate them. The global scheme consists of a set of OEM objects exported by wrappers to mediators. Mediators are specified using a language called Mediator Specificaion Language (MSL). Queries are expressed in MSL or in a specific language called LOREL (Lightweight Object Repository Language), an object-oriented extension of SQL. Each query is processed by a module, the Mediator Specification Interpreter (MSI).

It should be emphasized that TSIMMIS does not implement a real integration, as each mediator performs integration independently of each other. It means that does not exist the concept of a unified global scheme. The result of a query could be seen inconsistent and completely different from other mediators. This form of integration is called query-based.

GARLIC integration system is based on an architecture with Data Repositories at lowest level, which represent the data sources. Above each data repository we find a wrapper (called Repository Wrapper), which is responsible for communication between a data repository and the rest of the system. In addition, each wrapper ensures the transformation of the local schema of a source into a unified schema and transforming user queries into queries executable by data source.

The global schema has an object-oriented data model, managed by the Query Services and Runtime System components, and stored in the Metadata Repository, based on the

ODMG standard. ODMG objects are exported by wrappers using Garlic Data Language (GDL), based on the ODL (Object Definition Language) standard.

Unlike the TSIMMIS system, there is no mediator concept in GARLIC, and the integration of ODMG objects from different sources is performed by wrappers.

MOMIS (Mediator Environment for Multiple Information Sources) (*Orsini et al., 2009*; *Beneventano & Bergamaschi, 2004*) is a data integration system that manages structured and semistructured data sources. MOMIS is based on $I^3$ architecture (*Hull & King, 1995*), consisting of several wrappers and a mediator.

The integration methodology starts with an extraction activity where user uses a wrapper that transforms the structure of a data source into a $ODL_I3$ (Object Definition Language) model based on descriptive logic. The integration process generates an integrated view of data sources using global-as-view approach, building the global schema incrementally. At the end of the MOMIS integration process, starting when the query is posed by the user over the global schema, the mediator generates a $OQL_I3$ query and sends it to wrappers, which translate it into a query executable from the corresponding data source.

Ontology-based data access is by now a popular paradigm which has been developed in recent years to overcome the difficulties in accessing and integrating legacy data sources (*Xiao et al., 2018*). In OBDA, users are provided with a high level conceptual view of the data in the form of an ontology that encodes relevant domain knowledge. The concepts and roles of the ontology are associated via declarative mappings to SQL queries over the underlying relational data sources. Hence, user queries formulated over the ontology can be automatically rewritten, taking into account both ontology axioms and mappings, into SQL queries over the sources.

Overall, the large part of the analysed approaches, use their own description language, for both local and global schemas, and queries. However, if a generic external application wants to communicate with one of the systems presented, it should know the specific query language and/or the specific language used to describe the schemas. The problem of translation between languages is widened if we consider interoperability with the Web. For this reason, the proposed approach, Data Integration Framework (DIF), exploits the use of ontologies supported by a semiautomatic mapping strategy.

## DATA INTEGRATION FRAMEWORK

The proposed Data Integration Framework, is a generalization of both GaV and LaV approaches, based on data virtualization, and provides the possibility to define a mappings in both GaV approach (a correspondence between a view expressed in terms of the global schema and a view expressed in terms of the local schema) and LaV approach (correspondence between a view expressed in terms of the local schema and a view expressed in terms of the global schema). In addition, to overcome problems due to semantic heterogeneity and to support interoperability with external systems, ontologies are used as a conceptual schema to represent both data sources to be integrated and the global schema, and therefore each mapping is defined as a correspondence between elements of ontologies: concepts (or classes), datatype properties, and object properties. Since the

data virtualization approach is also used to define local ontologies, the construction of an ontology to represent a local source is guided by additional mappings, called source-mappings, defined as correspondences between elements of local ontology and elements that characterize the source itself (for example, for a relational source a mappings will be defined as a correspondence between an ontology concept and the table that represents it).

In the proposed solution, the query rewriting is used to reformulate a query posed over the global ontology into a set of queries posed over the local ontologies. This is due to the choice of using ontologies also to describe data sources to be integrated. In this way, though, the mediation process is not completed yet, since local ontologies do not contain real data. To complete the mediation process, a second query translation task is required to reformulate a query posed over the local ontology into a set of queries posed over the corresponding source.

**Definition 4.1** *(Data Integration Framework) The data integration framework DIF is a 5-uple $(O_g, O_l, A, MT, SML)$ where:*

- *$O_g$ is the global ontology, expressed in a $L_{O_g}$ logic.*
- *$O_l$ is the local ontology, expressed in a $L_{O_s}$ logic.*
- *A (Alignment) is a set of mappings $M_1, M_2, \ldots, M_n$ between ontologies $O_g$ and $O_l$. Each mapping $M_i$ is a 5-uple $(id, e_s, e_t, n, rel)$ where:*

  - *id is the unique mapping identifier;*
  - *$e_s$ and $e_t$, respectively, are the elements of the source ontology $O_s$ and target $O_t$. In the case of a GaV mapping type, $O_s$ represents the local ontology and $O_t$ the global one, vice versa in the case of a LaV mapping type;*
  - *n is a measure of confidence (typically within a range [0, 1]) that indicates the similarity between $e_s$ and $e_t$;*
  - *rel is a relationship between $e_s$ and $e_t$ (for example, equivalence, subsumption, disjunction).*

- *MT (Mapping Table) is a table whose rows represent an element $e_g$ of the global ontology $O_g$ and columns represent elements $e_{l1}, e_{l2}, \ldots, e_{ln}$ of the local ontology $O_l$ that are mapped to $e_g$.*
- *SML (Source Mapping List) is a set of mappings $SM_1, SM_2, \ldots, SM_n$ between the local ontology $O_l$ and the correspondent data source $S_i$. Each mapping $SM_i$, called source-mapping, is a triple $(id, src_k, dst_h)$ where:*

  - *id is the unique mapping identifier;*
  - *$src_k$ is a source element of the local ontology $O_l$.*
  - *$dst_h$ is a destination element of the local data source $S_i$ (for example, a table of a relational source).*

The framework must be able to handle both the integration process and the mediation process, which is shown in Fig. 4, making activities as automated as possible.

The integration process is divided into the following activities:

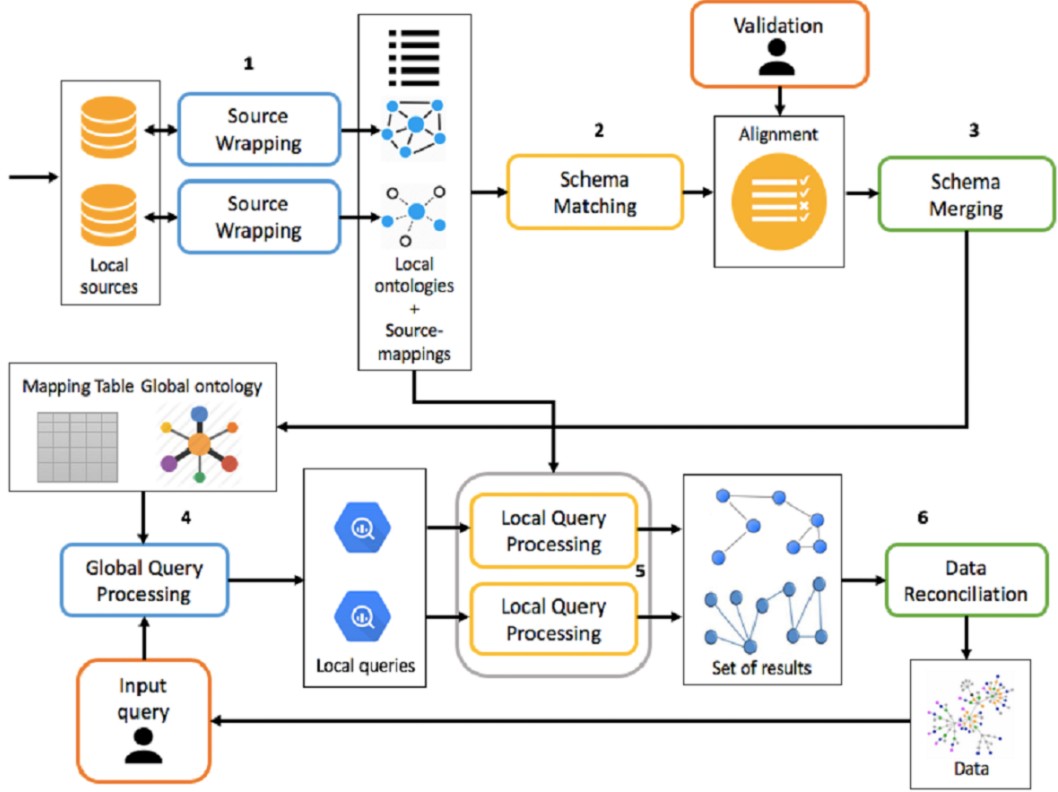

**Figure 4** Overview of integration and mediation processes.

1. Source Wrapping: for each source you want to integrate, you build an ontology to describe it. In addition, source-mappings are defined between the ontology and the data source, which will be subsequently used during the mediation process.
2. Schema Matching: for each local ontology, mappings are generated between it and global ontology. The matching activity generates mappings between a source ontology and a target one. Therefore, considering as target ontology the local one, it is possible to generate LaV mappings. Conversely, the followed approach will be GaV. Mappings are eventually validated or modified by the integrator designer. If the number of data sources to be integrated is 1, global and local ontologies are the same.
3. Schema Merging: each local ontology, taking into account the set of mappings defined in the previous activity, is integrated into the global ontology and the mapping table is updated. At this stage, global ontology is built incrementally.

   The mediation process, however, following a query submission, is divided into the following phases:

1. Global Query Processing: a query posed over the global ontology is reformulated, through rewriting, into a set of queries posed over local ontologies, using the mapping table generated at the end of the integration process;

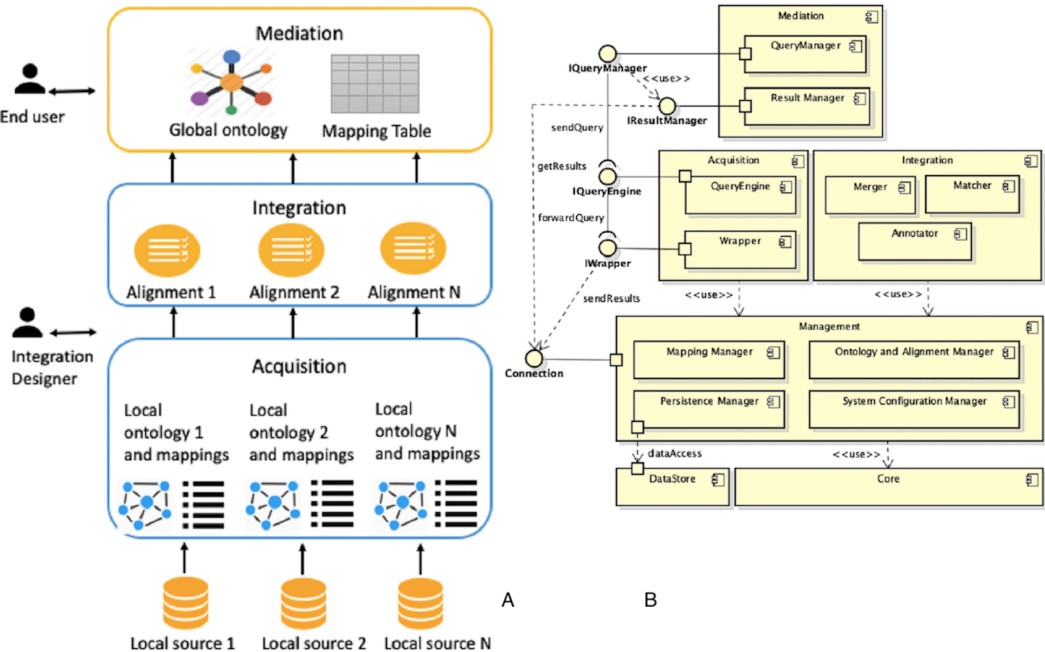

**Figure 5** Overview of DIF supporting tool: (A) integration approach and (B) UML diagram.

2.  Local Query Processing: each local query is reformulated into a set of queries over the corresponding data source, using source-mappings generated in the source wrapping activity. This set of queries, once executed, allows you to retrieve the real data.
3.  Data Reconciliation: extracted data from the previous activity is reconciled and combined before being presented to the user.

Local and global ontologies are expressed in OWL-DL (https://www.w3.org/TR/owl-features/), whose basic elements are classes $c$, object properties $op$ and datatype properties $dp$. Instances $i$ are not considered mapping because the data management approach is virtualization rather than materialization.

## DIF SUPPORTING TOOL

The tool, designed and developed to support the DIF framework, presents the typical architecture of integration systems based on the mediation approach (Fig. 5), providing two main components: mediator and wrapper.

According to Definition 4.1 and the description of the activities to be performed during integration and mediation processes, the architecture is composed by Acquisition, Integration and Mediation subsystems.

### Source wrapping

Data sources that the framework allows to integrate are structured and semi-structured (in particular, spreadsheet, JSON, and XML data sources). The source wrapping activity is performed by a class that implements the **IWrapper** interface. The output of this activity,

for each data source *S*, is a pair (*O*, *SML*) composed by the local ontology *O*, which describes the data source *S*, and the associated source mapping list *SML*.

### Relational data sources integration

The system allows the integration of relational data sources via JDBC connection (http://www.oracle.com/technetwork/java/javase/tech/index-jsp-136101.html) and supported databases are: MySQL, PostgreSQL, H2, DB2, Microsoft SQL Server, Oracle, Telid and MonetDB.

Relational data sources are connected to the framework by defining R2RML (https://www.w3.org/TR/r2rml/) mappings. Each R2RML mapping therefore represents a source-mapping, according with Definition 4.1. Local ontology is generated by identifying conditions associated to the database tables (*Ghawi & Cullot, 2007*) and, through identified conditions, associating each database element (table, column) to the corresponding ontology (class, datatype property, object property).

In addition to R2RML mapping, you can use a more compact notation, called axiom mapping, consisting of three elements:

- MappingID: mapping identifier;
- Source: a SQL query posed over the database;
- Target: RDF triple containing variables that refer to the names of the columns mentioned in the source query.

Each source source mapping $SM(id, src_k, dst_h)$ (Definition 4.1, contained in the source mapping list *SML*, contains an OWL resource (or local schema element) $src_k$ and an R2RML mapping (or an axiom mapping) $dts_h$.

### Spreadsheet data sources integration

Spreadsheet data sources are integrated with a new approach that seeks to examine the internal structure of the tables in order to extract an ontology that reflects as much as possible the data source. The approach is divided into several phases:

1. Source Scanning: the spreadsheet file is scanned in order to locate tables. At the end of the scan, a text string that describes the table structure is produced.
2. Parsing: the text string is parsed in order to generate the ontology elements, the relationships between them, and the physical location of cells within the spreadsheet data source. The output of this step is a list of schema attribute tables.
3. Analysis: an analysis of the list of attribute tables built to the previous step is performed to aggregate attributes with the same name in different concepts.
4. Restructuring: the generated ontology is refined in order to improve its quality.

Each source-mapping $SM(id, src_k, dst_h)$ contained the source mapping list *SML* (Definition 4.1) contains an OWL resource (or local schema element) $src_k$ and a data structure to track cells within the spreadsheet data source $dts_h$.

### XML data sources integration

XML data sources integration is based on its XSD schema (*Ghawi & Cullot, 2009*). If the XML schema does not exist, it is generated. Possible mappings are:

- Class mapping: XML nodes mapped as classes are complex types and element-group declarations (if they contain complex types). The XML schema supports two inheritance mechanisms (extensions and restriction), which are also mapped as classes.
- Property mapping: if an XML node has a simple type, or is an attribute-group declaration, or is an element-group declaration without additional associated complex types, it is mapped as properties, and its domain is the class corresponding to the parent node. Attributes are treated as simple types. In *Ghawi & Cullot (2009)*, instead, all element-groups and attributes-groups are mapped as classes.
- Relation mapping: if an element is a complex type and contains another element, whose type is also complex, a relationship is created between the respective classes.

The algorithm, in the ontology generation step, receives the XSG graph of the XML schema (XML Schema Graph) input. Starting from the root node, a deep visit is performed, generating an XPath expression for each visited node. Each source-mapping $SM(id, src_k, dst_h)$ contained the source mapping list $SML$ (Definition 4.1) contains an OWL resource (or local schema element) $src_k$ and the XPath expression $dts_h$.

## Schema maching

The goal of schema matching activity is to generate a set of mapping between local and global ontologies, which will then be validated by the user. The adopted approach generates mappings between classes, considering both semantic and contextual characteristics. Before to execute schema matching, a semantic annotation activity of ontologies is performed, whose output is a set of annotations $AN$, one for each $e_i$ element of the schema, where each annotation $AN_i$ is a triple $(tok_i, POS_i, sense_i)$ consisting of:

- $tok_i$: the token associated with the element $e_i$;
- $POS_i$: the lexical category of the token $tok_i$;
- $sense_i$: the meaning associated with $tok_i$ token for the lexical category $POS_i$, obtained as the output of the disambiguation process.

In the semantic matching task, a semantic-based matcher is applied to all pairs $(C_{Gi}, C_{Lj})$, where $C_{Gi}$ is the i-th class of the global schema, while $C_{Lj}$ is the j-th class of the local schema. The semantic matcher, for each pair $(C_G, C_L)$, generates the following information:

- *SemanticRel*: the type of semantic relation $(\equiv, \sqsubseteq, \sqsupseteq, idk)$;
- *SemanticSim*: is a coefficient $\in [0, 1]$ that specifies the reliability of the generated semantic relation.

Given $n$ and $m$ the number of local and global schema classes, respectively, the output of the semantic matching activity is:

$$(C_{Gi}, C_{Lj}) \underset{\substack{i \in [1, m] \\ j \in [1, n]}}{\Rightarrow} \begin{Bmatrix} SemanticRel \\ SemanticSim \end{Bmatrix}$$

The contextual matching activity generates mappings between classes taking into account how they are modeled, that is considering their properties. First, you must determine the equivalent properties between the two schemes. This is done by applying to all pairs

$(P_G, P_L)$, where $P_G$ and $P_L$ are the properties of the global and local schema respectively, the syntax-based or the semantic-based matcher. The syntax-based matcher, by analyzing the syntactic structure of words, returns for each pair $(P_G, P_L)$ a coefficient in a range $[0, 1]$. If the latter is greater than or equal to the $\beta$ threshold value, a mapping is generated between $P_G$ and $P_L$. The semantic-based matcher, instead, using WordNet to extract property sense, generates a mapping between $P_G$ and $P_L$ if there is an equivalence relation for at least one token pair. The semantic-based matcher is useful if synonyms are used to represent the same property and/or to discover mappings 1:n. Once the mappings have been discovered, it is possible to calculate, for all pairs $(P_G, P_L)$, the degree of contextual similarity, defined as *ContextualSim*, by applying the Jaccard measure:

$$ContextualSim(C_G, C_L) = \frac{|P(C_G) \cap P(C_L)|}{|P(C_G) \cup P(C_L)|}$$

where $P(C_G)$ and $P(C_L)$ are the set of properties of the classes $C_G$ and $C_L$ respectively.

The cardinality of the intersection of the two sets is equal to the number of existing mappings between the properties of the two classes. Given $n$ and $m$ the number of local and global schema classes, respectively, the output of the contextual matching activity is a set of pair as:

$$(C_{Gi}, C_{Lj}) \Rightarrow_{\substack{i \in [1, m] \\ j \in [1, n]}} \left\{ \begin{array}{c} ContextualSim \\ M_{P(C_{Gi}), P(C_{Lj})} \end{array} \right\}$$

where $M_{P(C_{Gi}), P(C_{Lj})}$ is the set of mappings between the properties of classes $C_{Gi}$ and $C_{Lj}$:

$$M_{P(C_{Gi}), P(C_{Lj})} = \left\{ \begin{array}{ccc} P_1(C_{Gi}) & \longleftrightarrow & P_1(C_{Lj}) \\ P_2(C_{Gi}) & \longleftrightarrow & P_2(C_{Lj}) \\ & \dots & \\ P_k(C_{Gi}) & \longleftrightarrow & P_z(C_{Lj}) \end{array} \right\}$$

where $k$ is the number of properties of the class $C_{Gi}$ and $z$ is the number of properties of the class $C_{Lj}$.

To determine which mappings can be returned to the user, a selection step is performed. The principle is that if there is a semantic relation *SemanticRel* and the degree of contextual similarity *ContextualSim* is greater than or equal to a threshold value, the corresponding mappings can be returned. By lowering these values, more weight is given to semantic characteristics rather than contextual ones. Given a semantic relation *Rel* and a threshold value $\alpha$, the algorithm selects 1:1, 1:n, n:1 and n:m mappings between all pairs of classes $(C_{Gi}, C_{Lj})$ if there is a semantic relation equal to *Rel* and if *ContextualSim* $\geq \alpha$. If more mappings 1:n, n:1 and n:m for the same pair of classes $(C_{Gi}, C_{Li})$ have a threshold value grater than $\alpha$, is returned the mapping with the largest number of classes. The output is the set of selected mappings. The output of schema matching activity, according to the Definition 4.1, is a alignment $A$ consisting of a set of mappings between the local and global

schema classes, obtained after the selection step:

$$A = \{M(\{C_G\}_k, \{C_L\}_z)\} \Rightarrow (C_{Gi}, C_{Lj}) \Rightarrow_{\substack{i \in [1,k] \\ j \in [1,z]}} \begin{Bmatrix} SemanticRel \\ Similarity \\ M_{P(C_{Gi}), P(C_{Lj})} \end{Bmatrix}$$

Mappings can be 1:1, 1:n, n:1 and n:m.

## Schema merging

The goal of schema merging activity, starting from user validated mappings, is the fusion between the local and global schema, generating a new virtual view. Schema merging activity is divided into two steps:

- In the first step, changes in the global schema are generated;
- In the second step, based on the proposed changes, the fusion of schemas is performed.

In the first step, given an input alignment $A$ (the mappings list), the global schema $G$ and the local one $L$, the new global schema $T$ is initially created, which is initially equal to $G$, and the empty mapping list $ML$ that will contain the mappings between $L$ and $T$ elements. Merging is performed by applying merge operators to each input mapping. Next, the local schema classes and relations, not included in the global schema, are added to $T$. The new resulting schema is modified by deleting redundant relationships and performing refactoring operations. The framework has an internal data structure to track changes to the new global schema $T$.

In the second step, given the changes produced and after deciding whether to validate or not, the real schema merging is performed.

Output of schema merging activity, besides to the new global schema $T$, according to the Definition 4.1, also consists of the mapping table $MT$, whose rows represent a $e_G$ element of the global schema $G$ and columns represent the $e_{L1}, e_{L2}, \ldots, e_{Ln}$ elements of the local schemas $L$ that are mapped to $e_G$. Since it is possible to define complex mappings n:m, the mapping table will be a table whose rows represent an $E_{Gi}$ expression of an element of the global schema $G$ and the columns represent the expressions $E_{Ljk}$ of the j-th element of the k-th local schema. A generic $MT[i,j]$ element of the mapping table represents, in fact, a mapping $M(id, E_{Gi}, E_{Ljk}, n, rel)$ between the expression of an element i-th of the global schema and an expression of an element j-th of the k-th local schema.

### *Mapping table*

In the mapping table, according with the Definition 4.1, rows represent elements of the global schema, and columns represent elements of the local schemes. Elements are generic OWL expressions and Table 2 shows the possible mappings in the mapping table:

The framework, however, allows mappings to be defined in a generic way, without explicit reference to a global or local schema. For this reason, the framework must be configured by setting a parameter $dir = \{global, local\}$ indicating the direction of the mappings in such a way as to support queries reformulation. If not specified, it is assumed that in the rows there are expressions referring to the global schema and in the columns

**Table 2  Mapping table.**

|  | Global schema $g$ | Local schemas $1, 2, \ldots, n$ |
|---|---|---|
| *CE* mapping | $CE_g$ | $\bigcup_{i \in [1,n]} CE_i$ |
| *OPE* mapping | $OPE_g$ | $\bigcup_{i \in [1,n]} OPE_i$ |
| *DPE* mapping | $DPE_g$ | $\bigcup_{i \in [1,n]} DPE_i$ |

the expressions referring to the local schemes. When it is necessary to insert a mapping in the opposite direction, it is inverted.

The mapping table is represented using the EDOAL (http://alignapi.gforge.inria.fr/edoal.html) language. For example, we consider the following mapping:

$$M(Hospital, Infirmary, \equiv, 0.4, [(Name, Name)])$$

For the mapping $M(Hospital, Infirmary)$, assuming to assign to the first schema the prefix *src#* and to the second schema the prefix *trg#*, the mapping representation for the property *Name* will be as follows:

```
<map>
<Cell>
  <entity1>
    <edoal:Property rdf:about="src#Name"/>
  </entity1>
  <entity2>
    <edoal:Property>
      <edoal:or rdf:parseType="Collection">
        <edoal:Property>
          <edoal:and rdf:parseType="Collection">
            <edoal:Property rdf:about="trg#Name"/>
            <edoal:PropertyDomainRestriction>
              <edoal:class>
                <edoal:Class rdf:about="trg#Infirmary"/>
              </edoal:class>
            </edoal:PropertyDomainRestriction>
          </edoal:and>
        </edoal:Property>
      </edoal:or>
    </edoal:Property>
  </entity2>
  <relation>=</relation>
  <measure rdf:datatype='http://www.w3.org/2001/XMLSchema#float'>0.4</measure>
</Cell>
</map>
```

## Query processing

The framework allows query execution by defining a query, posed over the global schema, through SPARQL (https://www.w3.org/TR/rdf-sparql-query/).

The query rewriting process (*Thiéblin et al., 2016*) exploits correspondences 1:n between global and local schema elements, expressed in descriptive logic, and applies a set of transformation rules to such correspondences.

Inputs of query rewriting process are a SPARQL query and a mapping table *MT* (in EDOAL format) and generates a set of queries, also expressed in SPARQL. Subsequently generated queries are transmitted to the acquisition subsystem for their evaluation, that is, to perform the local query processing task.

### Global query processing

Query rewriting process is performed by rewriting the graph pattern of a SPARQL query, applying the transformation rules to each triple pattern in it. Since a triple pattern can refer to data (for example, instance relationships) or schema (class and/or property relationships), or both, a pattern subdivision is performed based on the type. A triple pattern is a triple (*subject*, *predicate*, *object*), which can be classified as:

- DTP (Data Triple Pattern): if it is related to information concerning data and not the schema;
- STP (Schema Triple Pattern): if it is related to information concerning data and not the schema.

The reformulation process (Algorithm 1) applies the three-step transformation rules. In the first step, the triple is rewritten by considering the specified mappings for the *predicate* part. In the second step are considered mappings for the *object* part, and finally for the *subject* part. SPARQL variables, constants, and RDF/RDFS/OWL properties, which may appear in the subject, predicate, and object part of a triple, are not rewritten. As a result, the they will also appear in the rewritten query.

---

**Algorithm 1** SPARQL rewriting

---

**Input:** SPARQL query $Q_{in}$, mapping table $MT$
**Output:** SPARQL query $Q_{out}$

1:   $GP_{in} \leftarrow$ graph pattern of $Q_{in}$
2:   $GP_{out} \leftarrow GP_{in}$ after replacing IRIs in FILTER, using 1:1 mappings $MT$
3:   $GP_{out} \leftarrow$ TRIPLE PATTERN REWRITING($GP_{out}$, $MT$, *predicate*)
4:   $GP_{out} \leftarrow$ TRIPLE PATTERN REWRITING($GP_{out}$, $MT$, *object*)
5:   $GP_{out} \leftarrow$ TRIPLE PATTERN REWRITING($GP_{out}$, $MT$, *subject*)
6:   $Q_{out} \leftarrow$ new query containing $GP_{out}$

---

Transformation rules (*Thiéblin et al., 2016*) are described by a set of functions of the type:

$$D_y^x(t, \mu) \to TR \tag{1}$$

$$S_y^x(t, \mu) \to TR \tag{2}$$

where $t$ is a DTP (in Eq. (1)) or STP (in Eq. (2)), $\mu$ is the mapping between $e_s$ (source schema entity) and $e_t$ (target schema entity) for the subject, predicate or object part of $t$, $x \in \{s, p, o\}$ denotes the part of the triple used by the function, $y \in \{c, op, dp, *\}$ represents the type of $x$ (a class, relation, property or any, respectively) and $TR$ represents the transformation rule. A mapping $\mu$ is a generic element $MT[i, j] = M(id, E_{Gi}, E_{Ljk}, n, rel)$ of the mapping table $MT$. Although the mapping table allows managing 1:1, 1:n, n:1 and n:m mappings, the query reformulation process does not consider n:1 and n:m mappings. Functions Eqs. (1) and (2) are used to rewrite each triple of the input graph pattern. Output of global query processing is a set of queries, posed over the local ontologies, still expressed in SPARQL.

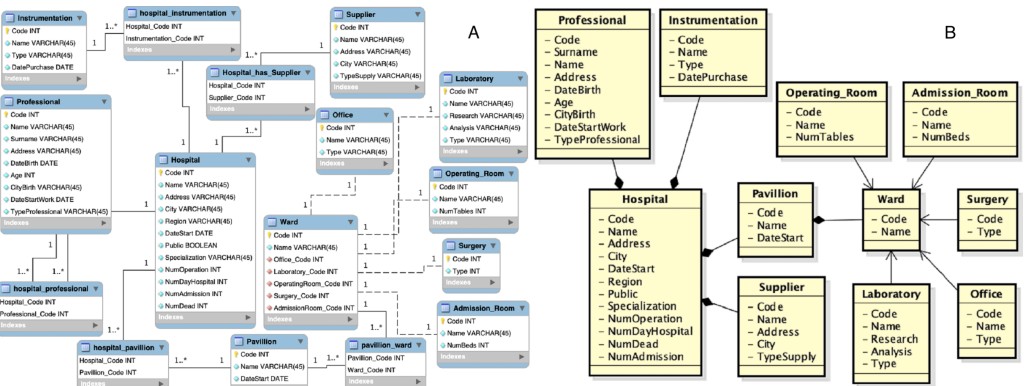

**Figure 6** First data source: (A) entity-relationship diagram and (B) local view.

### *Local query processing*

Local query processing is the second activity of the mediation process. Each reformulated query is still expressed in SPARQL and a second reformulation step is required for those data sources that use a language other than SPARQL to retrieve data.

Relational sources that the framework allows to integrate use SQL to express a query. To perform query reformulation, a SPARQL engine is used, which uses query rewriting techniques and inference mechanisms: Quest (*Rodriguez-Muro, Hardi & Calvanese, 2012*).

Query processing for XML data source is supported by a framework, integrated in the system, that allows query reformulation from SPARQL to XQuery (https://www.w3.org/TR/xquery-30/: SPARQL2XQuery (*Bikakis et al., 2009*).

## CASE STUDY

In order to validate the proposed framework, a case study was conducted using three heterogeneous data sources (two relational data sources, and one semi-structured, specifically a spreadsheet) designed in different contexts, related to the health domain applications.

As initial step the first source is acquired, which entity-relationship diagram and local view are shown in Fig. 6. At this point its local view becomes the new virtual view. In this case the only steps that must be performed are those of source wrapping and annotation of the schema. The extraction of semantic information, through the schema annotation activity, is necessary as this information will be used to generate the mapping with the source schemes that will be acquired later. The output of the schema annotation activity is a set of annotations $AN$, one for each element $e_i$ of the schema, where each annotation $AN_i$ is a triple $(tok_i, POS_i, sense_i)$ composed by:

- $tok_i$: the token of the element $e_i$;
- $POS_i$: the lexical category of the token $tok_i$;
- $sense_i$: the sense of the token $tok_i$ for the lexical category $POS_i$, as output of the disambiguation process.

Table 3 Output of the schema annotation activity.

| Class | Token | Sense |
|---|---|---|
| Hospital | Hospital | Sense#1: a health facility where patients receive treatment |
| Professional | Professional | Sense#2: an athlete who plays for pay |
| Supplier | Supplier | Sense#1: someone whose business is to supply a particular service or commodity |
| Instrumentation | Instrumentation | Sense#1: an artifact (or system of artifacts) that is instrumental in accomplishing some end |

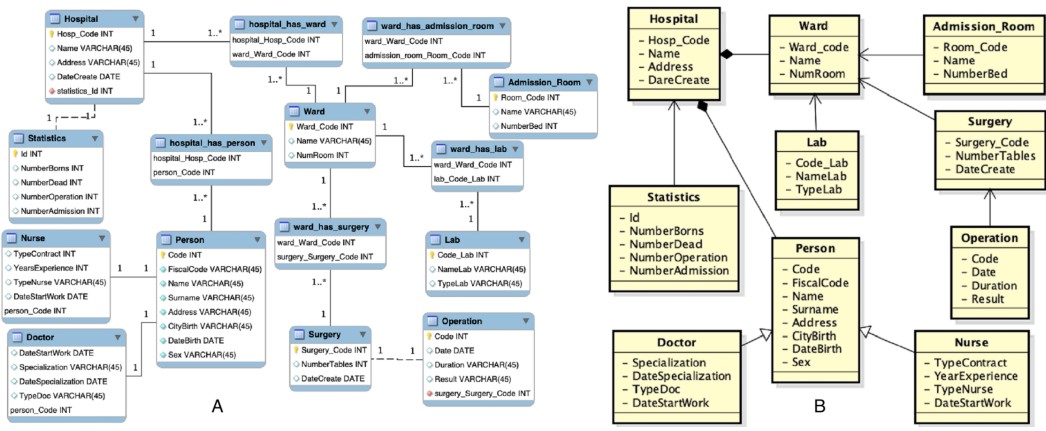

**Figure 7** Second data source: (A) entity-relationship diagram and (B) local view.

In Table 3 is shown an extracted of the output of the schema annotation activity performed over the first local view.

Then, the second source is acquired, which entity-relationship diagram and local view are shown in Fig. 7.

As in the first integration step, the source wrapping and schema annotation activities are performed. Subsequently, the schema matching activity is performed. To this aim, the following thresholds setting is adopted:

$$\alpha_{\equiv} = 0.2$$
$$\alpha_{\sqsubseteq/\sqsupseteq} = 0.3$$
$$\alpha_{idk} = 0.8$$
$$\beta = 0.8$$

Once the schema matching is completed the mappings are obtained. Some examples are:

$$M(Hospital, Hospital, \equiv, 1, [(Address, Address, 1), (Name, Name, 1), (Code, Hosp\_Code, 0.99)])$$

$$M(Oerating\_Room, Surgery, \sqsupseteq, 1, [(Code, Surgery\_Code, 0.99),$$
$$(NumTables, NumberTables, 0.84)])$$

$$M(Supplier, Person, \sqsubseteq, 1, [(Name, Name, 1), (Address, Address, 1),$$
$$(Code, Code, 1), (City, CityBirth, 0.99)])$$

$$M(Office, Lab, idk, 1, [(Type, TypeLab, 0.99), (Name, NameLab, 0.99),$$
$$(Code, Code\_Lab, 0.99)])$$

During the validation step of the mappings, the user should to delete the mapping $M(Office, Lab)$ and replace the semantic relationship of the mapping $M(Operating\_Room, Surgery)$ to ($\equiv$), as that relationship, in the SURGERY class of the first scheme, refers to a room in which a doctor can be consulted, while in the second scheme to an operating room. He also should to delete the correspondence between properties *City* and *CityBirth* in the mapping $M(Supplier, Person)$. The $M(Office, Lab)$ mapping is returned because the two classes match all properties and, as a result, the contextual similarity measure is 1. This mapping must be deleted otherwise during the schema merging activity a wrong association relationship will be created between the two classes. The $\alpha_{idk}$ threshold was chosen at 0.8 to highlight this observation. If association relationships have no reason to be created, the schema matching activity should be performed with a high value for the $\alpha_{idk}$ threshold. The threshold value $\alpha_{\sqsubseteq/\sqsupseteq}$ was chosen equal to 0.3 because, if it was lower, the mapping $M(Ward, Person)$ would be added a semantic relationship of hyponymy, but this mapping is wrong.

In the set of mappings should also to appear the mapping 1:n $M(Hospital, \{Hospital, Statistics\})$ but is not returned because of the threshold $\alpha_{idk}$ high. To take into account the representation of the hospital concept through the HOSPITAL and STATISTICS classes, there are therefore two alternatives. The first one is to keep the threshold value of $\alpha_{idk}$ high and insert manually the mapping. The second one is to lower the threshold and eliminate the other mappings in which there is a *idk* relationship, except the mapping except the mapping mentioned above. However, the following mapping is also obtained:

$M(Hospital, \{Hospital, Statistics\}) = [M(Hospital, Hospital), M(Hospital, Statistics)] = [$
$M(Hospital, Hospital, \equiv, 1, [$
$(Address, Address, 1), (Name, Name, 1),$
$(Code, Hosp\_Code, 0.99)]), M(Hospital, Statistics, idk, 0.21, [$
$(NumAdmission, NumberAdmission, 0.88), (NumDead, NumberDead, 0.80),$
$(NumOperation, NumberOperation, 0.88)])]$

The global view is initially the local view of the first source. At this point the schema merging activity is performed. The new global view is shown in Fig. 8. As an example, considering the mapping $M(Hospital, \{Hospital, Statistics\})$. Assuming the prefix *merged#* to the global view, the prefix *hospital_1#* to the first local view and *hospital_2#* to the second local view, once the schema merging activity is performed, the representation of the mapping $M(Hospital, \{Hospital, Statistics\})$, will be as follows:

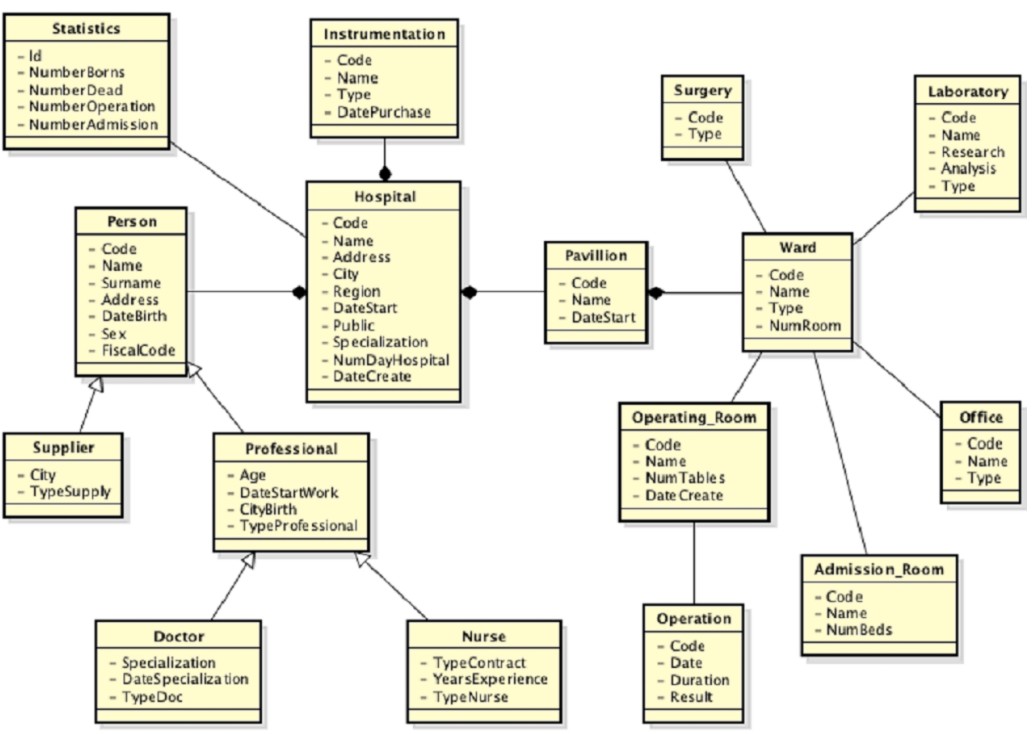

**Figure 8** Global view after the second source integration.

```
<map>
  <entity1>
    <edoal:Class rdf:about="merged#Hospital"/>
  </entity1>
  <entity2>
    <edoal:Class>
      <edoal:or rdf:parseType="Collection">
        <edoal:Class rdf:about="hospital_1#Hospital"/>
        <edoal:Class rdf:about="hospital_2#Hospital"/>
      </edoal:or>
    </edoal:Class>
  </entity2>
  <relation>=</relation>
</map>
<map>
  <entity1>
    <edoal:Class rdf:about="merged#Statistics"/>
  </entity1>
  <entity2>
    <edoal:Class>
      <edoal:or rdf:parseType="Collection">
        <edoal:Class rdf:about="hospital_2#Statistics"/>
      </edoal:or>
    </edoal:Class>
  </entity2>
  <relation>=</relation>
</map>
```

In a similar way the correspondences for the other elements of the schemes are defined.

**Figure 9** Third data source: (A) Part of spreadsheet 1 and (B) part of spreadsheet 2.

The third source acquired is a composed of different spreadsheets. Some parts of the spreadsheets are shown in Fig. 9.

An extracted of the local view of the third source is shown in Fig. 10.

After source wrapping and schema annotation activities are performed, the schema matching activity is performed using the following threshold values:

$$\alpha_{\equiv} = 0.2$$
$$\alpha_{\sqsubseteq/\sqsupseteq} = 0.3$$
$$\alpha_{idk} = 0.95$$
$$\beta = 0.1$$

The threshold value of the contextual similarity $\beta$ is equal to 0.1 because, although there are classes designed with different attributes, they represent the same concept of the real world and for which, therefore, the mappings must be returned. This situation is managed by lowering the value of $\beta$ but not those of $\alpha_{\sqsubseteq/\sqsupseteq}$ and $\alpha_{\equiv}$. The high value of $\alpha_{idk}$ is meant to filter almost all *idk* mappings, since they are not correct. It has been increased through tuning activities, in order to filter all those concepts with an empty set of mappings between their properties. In this way we can provide to the user just few mappings to be validated. An example of returned mappings to the user, are the following:

$M(Hospital, Hospital, \equiv, 1, [(Code, Hospital\_Code0.99), (Name, HospitalName, 0.99),$
$(Address, Street\_Address, 0.99), (City, City, 1)])$

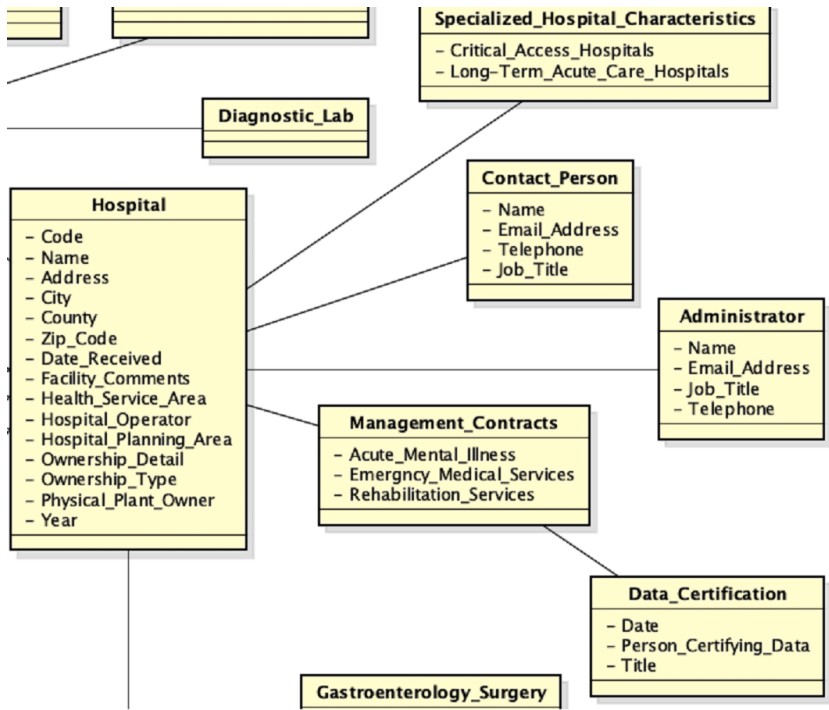

**Figure 10  Local view of the third source.**

$$M(Hospital, Rehabilitation\_Hospital, \equiv, 0.5, [(Code, Hospital\_Code, 0.99),$$
$$(Name, HospitalName, 0.99),$$
$$(Address, Street\_Address, 0.99), (City, City, 1)])$$

$$M(Hospital, Children\_Specialty_H ospital, \equiv, 0.33, [(Code, Hospital\_Code, 0.99),$$
$$(Name, HospitalName, 0.99),$$
$$(Address, Street\_Address, 0.99), (City, City, 1)])$$

$$M(Hospital, Psychiatric\_Hospital, \equiv, 0.5, [(Code, Hospital\_Code, 0.99),$$
$$(Name, HospitalName, 0.99), (Address, Street\_Address, 0.99), (City, City, 1)])$$

$$M(Person, Contact\_Person, \sqsupseteq, 0.5, [(Address, Email\_Address, 0.99), (Name, Name, 1)])$$

$$M(Person, Administrator, \sqsupseteq, 1, [(Address, Email\_Address, 0.99), (Name, Name, 1)])$$

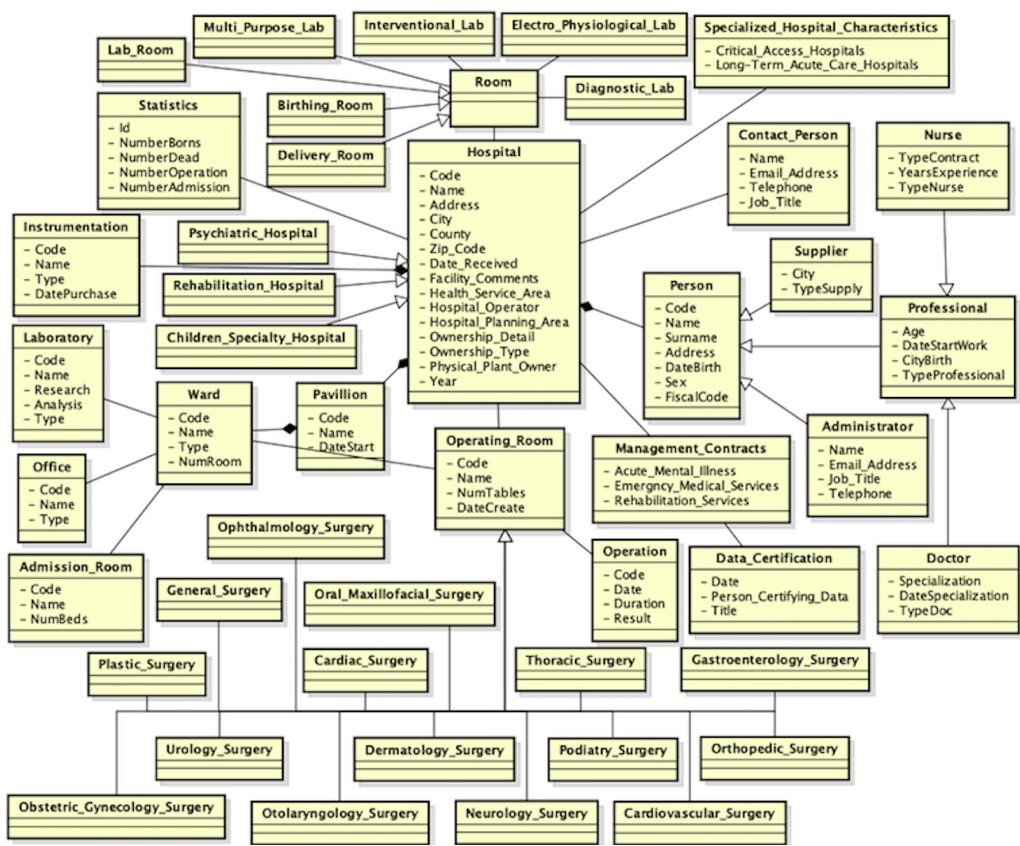

**Figure 11** New global view after the third data source is integrated.

During the validation step of the mappings, the user should to insert a *idk* mapping between the HOSPITAL and SPECIALIZED_HOSPITAL_CHARACTERISTICS classes, delete the correspondences between the *Address* and *Email_Address* properties in the mapping $M(Person, Administrator)$ and replace the semantic relationship of the mapping $M(Person, Contact\_Person)$ to *idk*. He also should to insert a mapping ≡ between the OPERATING _ROOM and SURGERY classes, as they both refer to an operating room. Besides, he should to replace the semantic relationship of the mappings $M(Hospital, Rehabilitation\_Hospital)$, $M(Hospital, Psychiatric\_Hospital)$ and $M(Hospital, Children\_Specialty_{H}ospital)$ to ⊒. Since the user has knowledge about the application domain, he is able to recognize which mappings must to be deleted or not. Once the schema matching activity has been completed, the next step is the schema merging activity. The new global view, in which all attributes are not shown, is shown in Fig. 11. After the third source is integrated, a lot of mappings are included, but many of them, as well as an example of the mapping table, are removed from the example, in order not to create confusion in the reader in understanding the full integration process.

## Query processing

In the query processing activity the user has the possibility to run a query over the global virtual view, through SPARQL (https://www.w3.org/TR/rdf-sparql-query/), as mentioned in 'Query processing'. We provide a short example of the query rewriting process, considering three queries. URIs used are *merged#* for the global virtual view and *hospital_1#*, *hospital_2#* e *hospital_3#*, respectively, for the first, second and third source.

The first query is: "Return all instances of the HOSPITAL class, with the corresponding names":

```
SELECT ?x ?y
WHERE {?x rdf:type merged#Hospital. ?x merged#Name ?y}
```

In this case, in the global view are merged *hospital_1#*, *hospital_2#* and *hospital_3#*, respectively, from the first, second and third local views. The mediation subsystem translates the above query in three queries, one for each of the integrated data sources. The reformulated queries are the followings:

```
SELECT   ?x ?y
WHERE
  { ?x   rdf:type hospital_1#Hospital.
      ?x hospital_1#Name   ?y ;
            rdf:type hospital_1#Hospital
  }

SELECT   ?x ?y
WHERE
  { ?x   rdf:type hospital_2#Hospital.
      ?x hospital_2#Name   ?y ;
            rdf:type hospital_2#Hospital
  }

SELECT   ?x ?y
WHERE
  { ?x   rdf:type hospital_3#Hospital.
     ?x   hospital_3#HospitalName   ?y ;
            rdf:type hospital_3#Hospital
  }
```

The second query is: "Return all instances of the PERSON class, with the corresponding names":

```
SELECT ?x ?name
WHERE { ?x rdf:type merged#Person. ?x merged#Name ?name}
```

The reformulated queries are the followings:

```
SELECT   ?x ?name
WHERE
  { ?x   rdf:type hospital_1#Professional
      { ?x hospital_1#Name   ?name ;
            rdf:type hospital_1#Supplier}
    UNION
      { ?x hospital_1#Name   ?name ;
            rdf:type hospital_1#Professional}
  }

SELECT   ?x ?name
WHERE
  { ?x rdf:type hospital_2#Person.
    ?x hospital_2#Name   ?name ;
       rdf:type hospital_2#Person
  }
```

**Table 4  Size of the local views.**

|  | First source | Second source | Third source |
|---|---|---|---|
| Number of classes | 11 | 10 | 33 |
| Number of relations | 15 | 11 | 5 |
| Number of properties | 25 | 35 | 27 |
| Number of instances | 340 | 280 | 220 |

```
SELECT  ?x ?name
WHERE
  { ?x rdf:type  hospital_3#Administrator.
      ?x hospital_3#Name  ?name ;
           rdf:type hospital_3#Administrator
  }
```

The third query is: "Return all instances of the PERSON class living in Benevento":

```
SELECT ?person ?city
  WHERE { ?person rdf:type merged#Person. ?person merged#Address ?city
    FILTER regex(?city, "Benevento", "i") }
```

The reformulated queries are the followings:

```
SELECT  ?person ?city
WHERE
  { { ?person rdf:type hospital_1#Professional
    { ?person hospital_1#Address ?city ; rdf:type hospital_1#Professional}
      UNION
    { ?person hospital_1#Address ?city ; rdf:type hospital_1#Hospital}
      UNION
    { ?person hospital_1#Address ?city ; rdf:type hospital_1#Supplier}
    }
    FILTER regex(?city, "Benevento", "i")
  }

SELECT  ?person ?city
WHERE
  { { ?person rdf:type hospital_2#Person
    { ?person hospital_2#Address ?city ; rdf:type hospital_2#Person}
      UNION
    { ?person hospital_2#Address ?city ; rdf:type hospital_2#Hospital}
    }
    FILTER regex(?city, "Benevento", "i")
  }

SELECT  ?person ?city
WHERE
  { ?person rdf:type hospital_3#Administrator ;
    hospital_3#Street_Address  ?city ; rdf:type hospital_3#Hospital
    FILTER regex(?city, "Benevento", "i")
  }
```

If an element does not have a correspondence with an element of the some local view, the translated query for that view is the same of the global view. Each wrapper will return an empty result when the query will be performed.

## Analysis

We report time overheads in each of the phases of the approach proposed. In the case study presented is shown the application of the approach rather than optimizations of the performance of the activities of the integration process. For this reason, as shown in Table 4, the size of the data sources, in terms of the number of the elements of the structures that represent them, is not high. Nevertheless, the developed software prototype shows good

**Table 5  Time overheads of the proposed approach.**

| Activity | Time (ms) |
|---|---|
| Source wrapping (first source) | 166 |
| Source wrapping (second source) | 87 |
| Source wrapping (third source) | 552 |
| Schema matching (first and second views) | 462 |
| Schema matching (global and third views) | 616 |
| Schema merging (first and second views) | 71 |
| Schema merging (global and third views) | 85 |
| Total time of the integration process | 2,039 |

**Table 6  Time overheads of the query processing activity.**

| | First query | Second query | Third query |
|---|---|---|---|
| Query rewriting time (ms) | 255 | 220 | 207 |
| Query execution time (ms) | 2,007 | 2,002 | 2,229 |

performance in terms of the execution time of the proposed approach phases, as shown in Table 5. The acquisition of Excel data sources has a longer execution time than acquiring relational data sources. This is because we need to consider the access times to the file and the identification of the tables that will constitute the elements of the local view. The low execution times of the schema matching and merging activities are relatively low, as there are optimizations of the algorithms and data structures used. To the total execution time of the full integration process, the time necessary to validate the mappings, which depends on the user, and the setup time needed for the schema matching activity (about 6 s) must be added. During the setup of the schema matching activity, performed only once, the modules needed for the annotation activity of the local views are loaded. Table 6, instead, shows the execution times of the query processing activity. The transformation of the queries has low execution times because the prototype is supported by the mapping table. With the mapping table we can reduce the time for searching an element (class, property or relationship) inside the local view that should be replaced in the query. This is not true when the query is really executed, because the time of execution depends on the specific technology of a data source.

## CONCLUSIONS

The purpose of this paper is to allow unified access to heterogeneous and independent source data, offering a data integration approach that addresses all the issues discussed. The architecture adopted is that of mediation systems, which create a virtual view of the real data and allow to external applications to access data through that view in a transparent manner. Transparency is guaranteed by translating queries posed over the virtual view into queries that are directly executable from local sources.

The proposed approach allows unified access to heterogeneous sources through the following activities:

- Source wrapping: the initial activity is the construction of an ontology for each source you want to integrate, whose structure is subsequently refined by using information extraction techniques to improve the quality of ontology.
- Schema matching: ontologies are then put in a matching process in order to automatically search mappings between the elements of the structures, using both syntax-based and semantic-based techniques. Mappings are identified by combining both semantic and contextual characteristics of each element. These mappings are then validated and, if necessary, modified by the user.
- Schema merging: based on the generated mappings, a global ontology is created which is the virtual view of the system.
- Query reformulation: at this stage, a query posed over the virtual view is reformulated into a set of queries directly executable by local sources. The reformulation task is performed automatically, generating an execution plan of the reformulated queries, with the possibility for the user to modify each single query.

Overall, the approach is semi-automatic, but compared to existing systems, the user's effort is minimized as he only intervenes in the matching configuration activity, by setting the threshold values for the mappings generation, and mappings validation. Both simple (1:1) and complex mappings (1:n, n:1 and n:m) are generated.

The outlined approach is supported by a specially designed and developed software system. The system provides a first level of abstraction of the activities and components involved in their execution and a second level of component specialization. Although the design of the system is aimed at covering all aspects of data integration described so far, implementation has some limitations. In particular, the acquisition of unstructured sources is not yet contemplated in development and the data reconciliation process requires the development of appropriate components. Except for such activities, integration and mediation processes are fully supported by the system.

Research activities that will be carried out in the future will have the goal of overcoming the limitations shown and consolidating, at the same time, the part of the system developed so far. In particular, accurate experimentation is required for validating the proposed approach, for ensuring high quality of mappings and local and global views, for optimizing the mediation process.

### Funding
The authors received no funding for this work.

### Competing Interests
The authors declare there are no competing interests.

## Author Contributions

- Giuseppe Fusco and Lerina Aversano conceived and designed the experiments, performed the experiments, analyzed the data, performed the computation work, prepared figures and/or tables, authored or reviewed drafts of the paper, and approved the final draft.

## Data Availability

Data and code are available at GitHub: https://github.com/gppfusco/DataIntegrationFramework.

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
