# Peer review of "An approach for semantic integration of heterogeneous data sources"

_PeerJ Computer Science, doi:10.7717/peerj-cs.254_

## Round 0.1 · original submission · Major Revisions

In general, both reviewers are favorable to accepting the paper after some revisions, so read carefully their recommendations and prepare a new version of the manuscript addressing all of them

Reviewer 1 ·

Basic reporting

The introduction and related work sections contain excessive background. As explaining concepts like heterogeneity and mappings are required, I would suggest a specific section for previous concepts, so that the related work is focused on existing integration approaches only. With less concepts, the introduction would better highlight the context, motivation and contribution of the paper.

Similarly, the abstract should be extended to describe the key characteristics of the approach, e.g. global and local views and the use of ontologies, and how such a solution constitutes a progress in the field.

The text of some figures (e.g. Figures 5 and 6) is difficult to read. Please increase size or resolution. Figures should be located close to the text describing them (e.g. Figures 4 and 5).

Table 2 is not referenced in the text.

Avoid the use of contractions (don’t)

Experimental design

The paper does not formulate neither explicit nor implicit research question. The introduction should formally state the research problem and briefly mention how the proposed system and the experiments effectively solve it.

The methods for relational data sources integration, i.e. R2RML and axiom mappings, should be explained in more detail. Similarly, the annotation activity is mentioned as a necessary step in the case study, but I cannot find a subsection devoted to it in section 4.

Validity of the findings

The authors present a case study, which is valuable, but from which general conclusions are more difficult to extract. Besides, some of the decisions taken during the example transformations are not justified: 1) how thresholds are set and why they change when a third source is included (only the value of beta is mentioned); 2) some mappings are manually removing, but it would be difficult for the user to know when or why these relations should be discarded. Due to the focus on a case study, the authors should go into the details. Some guidelines in this regard would be highly appreciated.

Additional questions regarding the experimentation are: 1) how the system solve or detect a query that cannot be translated into one of the sources? E.g. the local ontology does not contain the information asked by the user; and 2) Time overhead (or estimations) in each of the phases of the approach could be reported, since the approach performs multiple transformations. Also, scalability tests could be planned to study the influence of the number of instances in more realistic contexts.

Additional comments

The paper presents a data integration system for heterogeneous data sources that combines the benefits of the two traditional approaches, namely global-as-view and local-as-view. The approach is based on the use of ontologies at both global and local levels. Mediator elements adapt queries from the global view to the local views. After describing the components of the architecture, a case study in the area of medical information is presented.

The paper provides a solid background on data integration and its related issues, but at the cost of late introduction to the scientific contribution. Many technical solutions might exist for data schema and mediation, so the authors should clearly motivate why there is a scientific problem here and why ontology-based solutions are the most suitable way to address it.

A fully operative system implementing the approach seems not to be available yet. Although it should not be a problem to report inconclusive results in PeerJ, I encourage the authors to provide access to some prototype or controlled example to reach a wider audience.

Reviewer 2 ·

Basic reporting

The manuscript is overall well written and it has merit for publication

Experimental design

There is no technically a experimental evaluation but a presentation of a case study.

Validity of the findings

Better comparison with related approaches would be required

Additional comments

The abstract is too short and simplistic.
Authors should extend it and highlight the contribution of this paper.
Authors should avoid the excessively use of bullet points in the manuscript.
There is a broad but not too deep coverage of the state of the art and related technologies, a more in depth analysis of existing approaches and clarifying how the proposed methodology overcomes the limitations of the state of the art would be appreciated.

---

## Round 0.2 · Minor Revisions

Please follow the recommendation of reviewer 1. The paper is almost ready for publication.

Reviewer 1 ·

Basic reporting

The introduction has been greatly improved. The separation between background and related works makes reading easier. The only issue here is subsection 3.1, as no other subsections appear in section 3. Besides, the discussion is focused on a comparison between GaV and LaV, not specific approaches taken from the literature, so maybe it would be more useful at the end of section 2.

The quality of the figures is better now. However, there are missing references to figures in lines 448 and 473 (section 6). Please check that figures/tables are always written with capital letter at the beginning or fully lowercase (see journal format).

I strongly recommend proof-reading. I have found several typos, such as:
is an hard task => a hard task
It exploit => exploits
is ho to => how

Experimental design

The introduction now refers to the research problem and proposed solution.

The explanation of the method has been properly extended.

Validity of the findings

The authors have made changes to address my comments.

Additional comments

The authors have made important efforts to improve the paper. The contribution and reporting of findings are now better explained. I only have some minor comments for authors to fix before publication.

Reviewer 2 ·

Basic reporting

no further comments

Experimental design

no further comments

Validity of the findings

no further comments

Additional comments

Authors have addressed the comments of the reviewers

---

## Author Rebuttal · Round 0.2

Dear editor and reviewers,

this submission is the revised version of the paper entitled "An Approach for Semantic Integration of Heterogeneous Data Sources", which was recommended as "Major Revisions".

Firstly, we sincerely appreciate editor and reviewers' efforts for helping us to improve the work, with their kind and valuable comments.

The following of this document lists the responses to the comments. *The sentences in italic type, are reviewers' comments*; while **the sentences in bold type are the authors' responses**.

Overall, we sincerely appreciate the comments received for our work!

Best regards!
Yours Sincerely,
Lerina Aversano
Giuseppe Fusco

November 13, 2019

# Responses:

We carefully revised our paper based on the comments received.

Reviewer #1

*Basic reporting*

*The introduction and related work sections contain excessive background. As explaining concepts like heterogeneity and mappings are required, I would suggest a specific section for previous concepts, so that the related work is focused on existing integration approaches only. With less concepts, the introduction would better highlight the context, motivation and contribution of the paper.*

**Response: thanks for the suggestion. The related work section has been carefully reviewed and reorganized in two sections. Moreover, the introduction has been revised to better highlight the context and authors contribution.**

*Similarly, the abstract should be extended to describe the key characteristics of the approach, e.g. global and local views and the use of ontologies, and how such a solution constitutes a progress in the field.*

**Response: The abstract has been carefully extended and now it includes the main characteristics of the approach. Thank for the comment.**

*The text of some figures (e.g. Figures 5 and 6) is difficult to read. Please increase size or resolution. Figures should be located close to the text describing them (e.g. Figures 4 and 5).*

**Response: done.**

*Table 2 is not referenced in the text.*

**Response: Due to the restructuring of the section table 2 has been deleted.**

*Avoid the use of contractions (don't)*

**Response: done.**

*Experimental design*

*The paper does not formulate neither explicit nor implicit research question. The introduction should formally state the research problem and briefly mention how the proposed system and the*

*experiments effectively solve it.*

**Response: Thank for the comment. Now the research problem addresses in the paper is clearly introduced in the introduction. Moreover, the solution of the proposed approaches is briefly outlined.**

*The methods for relational data sources integration, i.e. R2RML and axiom mappings, should be explained in more detail. Similarly, the annotation activity is mentioned as a necessary step in the case study, but I cannot find a subsection devoted to it in section 4.*

**Response: The description of the methods has been extended, and the annotation activity explained in subsection "5.2 Schema maching".**

*Validity of the findings*

*The authors present a case study, which is valuable, but from which general conclusions are more difficult to extract. Besides, some of the decisions taken during the example transformations are not justified: 1) how thresholds are set and why they change when a third source is included (only the value of beta is mentioned); 2) some mappings are manually removing, but it would be difficult for the user to know when or why these relations should be discarded. Due to the focus on a case study, the authors should go into the details. Some guidelines in this regard would be highly appreciated.*

**Response: thanks for the comment. The value of the threshold idk has been justified. The second threshold that has been changed it was an error during system configuration. Besides, the section has been extended.**

*Additional questions regarding the experimentation are: 1) how the system solve or detect a query that cannot be translated into one of the sources? E.g. the local ontology does not contain the information asked by the user; and 2) Time overhead (or estimations) in each of the phases of the approach could be reported, since the approach performs multiple transformations. Also, scalability tests could be planned to study the influence of the number of instances in more realistic contexts.*

**Response: thanks for the comment. We have extended the section about the query execution and reported the time overhead for each phase of the approach. The latter in a separate section. The scalability tests are planned but not yet executed as the main effort was committed to find solutions to overcome some limitations of the previous version of the prototype.**

*Comments for the author*

*The paper presents a data integration system for heterogeneous data sources that combines the benefits of the two traditional approaches, namely global-as-view and local-as-view. The approach is based on the use of ontologies at both global and local levels. Mediator elements adapt queries from the global view to the local views. After describing the components of the architecture, a case study in the area of medical information is presented.*

*The paper provides a solid background on data integration and its related issues, but at the cost of late introduction to the scientific contribution. Many technical solutions might exist for data schema and mediation, so the authors should clearly motivate why there is a scientific problem here and why ontology-based solutions are the most suitable way to address it.*

*A fully operative system implementing the approach seems not to be available yet. Although it should not be a problem to report inconclusive results in PeerJ, I encourage the authors to provide access to some prototype or controlled example to reach a wider audience.*

**Response: thank you for your valuable comments. We have carefully reviewed the paper trying to satisfy all the suggestions received.**

#Reviewer 2

*Basic reporting*

*The manuscript is overall well written and it has merit for publication*

**Response: thanks for the positive evaluation of the paper.**

*Experimental design*

*There is no technically a experimental evaluation but a presentation of a case study.*

**Response: yes, that is correct the paper reports an evaluation performed through a case study.**

*Validity of the findings*

*Better comparison with related approaches would be required*

**Response: Thanks for the comment. The comparison has been extended.**

*Comments for the author*

*The abstract is too short and simplistic.*

*Authors should extend it and highlight the contribution of this paper.*

*Authors should avoid the excessively use of bullet points in the manuscript.*

*There is a broad but not too deep coverage of the state of the art and related technologies, a more in depth analysis of existing approaches and clarifying how the proposed methodology overcomes the limitations of the state of the art would be appreciated.*

**Response: thank you. Both the abstract and the related work sections have been carefully improved.**

---

## Round 0.3 · accepted · Accept

After reviewing the new version of the manuscript and the changes based on the suggestions given by reviewers, I think the paper is ready for publication now.